# Review of Application of Machine Learning as a Screening Tool for Diagnosis of Obstructive Sleep Apnea

**DOI:** 10.3390/medicina58111574

**Published:** 2022-11-01

**Authors:** Ishan Aiyer, Likhita Shaik, Alaa Sheta, Salim Surani

**Affiliations:** 1Blair Academy, Blairstown, NJ 07825, USA; 2Department of Medicine, Hennepin Healthcare, Minneapolis, MN 55404, USA; 3Department of Computer Science, Southern Connecticut University, New Haven, CT 06515, USA; 4Department of Medicine, Texas A&M University, College Station, TX 77843, USA

**Keywords:** sleep apnea, ML, screening, artificial intelligence

## Abstract

Obstructive sleep apnea syndrome (OSAS) is a pervasive disorder with an incidence estimated at 5–14 percent among adults aged 30–70 years. It carries significant morbidity and mortality risk from cardiovascular disease, including ischemic heart disease, atrial fibrillation, and cerebrovascular disease, and risks related to excessive daytime sleepiness. The gold standard for diagnosis of OSAS is the polysomnography (PSG) test which requires overnight evaluation in a sleep laboratory and expensive infrastructure, which renders it unsuitable for mass screening and diagnosis. Alternatives such as home sleep testing need patients to wear diagnostic instruments overnight, but accuracy continues to be suboptimal while access continues to be a barrier for many. Hence, there is a continued significant underdiagnosis and under-recognition of sleep apnea in the community, with at least one study suggesting that 80–90% of middle-aged adults with moderate to severe sleep apnea remain undiagnosed. Recently, we have seen a surge in applications of artificial intelligence and neural networks in healthcare diagnostics. Several studies have attempted to examine its application in the diagnosis of OSAS. Signals included in data analytics include Electrocardiogram (ECG), photo-pletysmography (PPG), peripheral oxygen saturation (SpO2), and audio signals. A different approach is to study the application of machine learning to use demographic and standard clinical variables and physical findings to try and synthesize predictive models with high accuracy in assisting in the triage of high-risk patients for sleep testing. The current paper will review this latter approach and identify knowledge gaps that may serve as potential avenues for future research.

## 1. Introduction

Obstructive sleep apnea (OSA) is a common chronic medical disease with an estimated prevalence rate of 5–14% [1]. A recent study has suggested that the actual prevalence of OSA could be much higher, with estimates ranging from 24% to as high as 50% in the mean which would make it the most prevalent chronic medical condition. Besides high prevalence, OSA has a wide-ranging impact on health and has been shown to be associated with an increased risk of hypertension, cardiovascular disease, cerebrovascular disease, atrial fibrillation, impaired glycemic control, erectile dysfunction, and gastroesophageal reflux, as among other conditions. Moreover, sleep disruption caused by sleep apnea leads to significant daytime sleepiness with implications for risk for motor vehicle accidents, especially in high-risk employees such as truck drivers or those heavy operating machinery. It can also lead to significant changes in mood, including depression, and cause attention deficit and loss of memory [2].

Traditionally the diagnosis of sleep apnea has been made by overnight polysomnography, which is done in a sleep laboratory. However, this requires significant investment in infrastructure, including hiring certified personnel, and hence became an impractical solution to diagnose sleep apnea efficiently. With the advent of home sleep testing, it became easier to screen more people for sleep apnea. Still, given the high disease prevalence, and relatively small numbers of home sleep testing equipment in use, especially among certified sleep providers, this approach is associated with significant barriers and diagnosis delays.

It has become increasingly apparent that we need a better strategy to accurately identify those at high risk for sleep apnea to use testing devices more efficiently. Previously, physicians relied on questionnaires such as the Epworth sleepiness scale (ESS) or the STOP-BANG questionnaire to screen for patients and triage those with high risk for sleep testing. However, this has increased reliance on patient self-reporting of symptoms. In the past few years, there has been an emergence of predictive models that take into account demographic and anthropometric data and the presence of comorbidities and symptoms to improve the accuracy of screening [3,4,5,6].

With the explosive growth of machine learning (ML) and artificial intelligence and its increasing use in healthcare, recently, the focus has turned to applications of ML in developing predictive models to improve the screening strategy for OSA and increase its sensitivity and accuracy. This article will highlight some of the critical studies and reports that cover this topic in the recent literature.

## 2. Why Machine Learning?

The diagnostic approach towards OSA is based off symptomatology and clinical suspicion leading to PSG study that is a gold standard for OSA diagnosis. However, the entire process is time taking and inconvenient and burdensome for patients. Moreover, the American Academy of Sleep medicine (AASM) has stated that the diagnosis of OSA should involve more reliable systematic diagnostic methods, than only through apnea-hypopnea index (AHI). For this purpose, ML is an evolving technology that combines pre-determined OSA indicative aspects to develop a tool to diagnose future patients. Researchers over decades have devised various methods to shape automated tools to diagnose OSA [3,4].

ML (ML) is a sub-field of artificial intelligence that describes how computers create pattern recognition and the capacity to learn from examples and make predictions based on historical data without being explicitly programmed to do so. ML provides highly complex algorithms. The way these algorithms work depends on the type of application. The whole idea depends on creating a model that can learn from data. Any task based on data points can be programmed using ML; even more, complex tasks like understanding email spam, customer interest, intelligent car driving, and classifying objects.

The four major ML models are supervised, unsupervised, semi-supervised, and reinforcement. With supervised learning, the model uses a dataset with data and we provided labels (classifications of the observations. Supervised models can be further classified based on output variables into regression and classification—in regression output variables have a real continuous value like weight, while in classification the output variable is a category like obese/not obese.

In unsupervised learning, model works with a dataset without any labels and is then able to explore the data to infer a function to describe relationships from the unlabeled data. The system does not predict the right output but is used to detect clustering (inherent grouping based on shared traits) or association rules (association between different characteristics). Semi supervised models are a combination of both above where the model uses labeled as well as non-labeled data. Reinforcement models involve a sequence of decisions made based on error and reward with a goal of maximizing reward

In Figure 1, we show the four major ML models.

## 3. Machine Learning in Screening for OSA

For the purpose of this review, we searched for relevant articles using PubMed and only included those studies that described applications of ML for screening for OSA using clinical features. We specifically excluded studies where ML was primarily applied to automate data from monitoring devices such as oximetry, or electrocardiogram or was used to study data acquired during sleep testing or polysomnography. We presented a summary of the papers that were included in this review in Table 1 [5,6,7,8,9,10,11,12,13,14]. Kirby et al. used a neural network to train using 23 clinical variables in 255 patients, and subsequently evaluated the predictive properties in a cohort of 150 additional patients. OSA prevalence was 69% for the sample. The trained generalized regression neural network (GRNN) had an accuracy of 91.3% (95% confidence interval (CI), 86.8 to 95.8). Values for sensitivity was 98.9% for having OSA (95% CI, 96.7 to 100), and specificity was 80% (95% CI, 70 to 90). Positive predictive value was high at 88.1% (95% CI, 81.8 to 94.4), as was the negative predictive value of 98% (95% CI, 94 to 100). This approach of applying Artificial neural networks (ANNs)to the clinically based prediction of OSA allows involving a more variables than utilized in linear or logistic regression techniques [11,15].

Mencar et al. selected a mixture of respiratory signals and clinical variables to include 19 variables with a communality index ≥0.50 out of the 32 initial features to train classification models and regression models to evaluate the prediction OSA severity ability of represented either by class or by available apnea–hypopnea indices from PSG. The discriminating power between normal subjects and sufferers from OSAS has been examined considering different efficacy/efficiency parameters such as classification accuracy (CA), precision, sensitivity, and Area under curve (AUC) by means of cross validation. The Support vector machine (SVM) classification model, trained with the first eight features gave the best results in terms of CA (44.7%) and AUC (65%). In terms of precision/recall (44.1%), the random forests (RF) classification model trained with the first five features showed the best results. However, these resulted need to be validated in larger populations with wider set of comorbidities. This study identified only body mass index (BMI) as a single factor that could screen for OSA and also predict the severity [5]. In a study of 2690 patients, Bouloukaki et al. NC was seen to be the best correlate with the AHI as a predictor of OSA [16].

Holfinger et al. used data from international Sleep Apnea Global Interdisciplinary Consortium (SAGIC) (*n* = 1613) and the Sleep Heart Health Study (SHHS) (*n* = 5599) to compare logistic regression and ML techniques, including artificial neural network (ANN), RF, and kernel SVM. They looked at demographic features including age, gender, ethnicity to predict OSA. A cohort of 17,448 subjects was randomly divided into two sets- the first training set had 10,469 patients and a second group which was the validation set included6,979 patients. The AUCs (95% CI) of the ML models (ANN, 0.68 (0.66–0.69) were significantly higher than logistic regression (0.61 (0.60–0.62)) in both the training and validation datasets. In both the SAGIC testing sample, and the SHHS testing sample, the ANN had AUCs similar to those of STOP-BANG [12].

Ustun et al. used 3 subsets of features (Electronic Medical Records (EMR) extractable “size 5”, EMR extractable “size 10” and self-self-reported symptoms) to train the Supersparse Linear Integer Model (SLIM) and the classification models. SLIM and classification models were trained at 19 and 39 different points on the receiver operating characteristics (ROC). For each model class they specifically ran SLIM to produce a model that had the highest true positive rate (TPR) subject to a constraint on the maximum allowable false positive rate (FPR) ≤ 5%, 10%, and so on. The performance of models using only extractable features sometimes exceeded the performance of models using all features at some points of the ROC curve, possibly since using a smaller set of features prevents over-fitting. The predictive utility of extractable features was superior to that of symptom-based features, regardless of the classification method used. The posttest probability of the SLIM model also varied with the prevalence of OSA (Table 1) [6].

In a study from Taiwan on 120 patients undergoing overnight PSG the intake questionnaire was analyzed making use of a genetic algorithm (GA) in order to build the five best models, and analysis using logistic regression (LR) served as a comparator. Sensitivity (81–88%) and specificity (95–97%) of the GA model was seen to be much higher compared to the LR model (55.6% and 57.9%, respectively) [13]. Zhang et al. showed that a SABIHC2 (Sex-Age-Body mass index (BMI)-maximum Interincisal distance-ratio of Height to thyrosternum distance-neck Circumference-waist Circumference) machine-learning model provides a simple and accurate assessment of moderate to severe OSA in the Chinese population. For this study, they set up a SABIHC2 model. The SABIHC2 model was able to screen for OSA better than STOP-BANG (AUC = 0.83 vs. 0.62, sensitivity 0.92 vs. 0.49 specificity 0.75 vs. 0.77 respectively) Interestingly the model performed better for asymptomatic patients (ESS < 10) [8].

Huang et al. developed a SVM model incorporating features from routinely collected parameters during clinical evaluation from 6875 Chinese patients referred for suspected sleep apnea. AHI cutoffs of 5, 15 and 30 were used to stratify OSA severity. The modeling was achieved through fivefold cross-validation. Two features were selected to predict AHI cutoff ≥5/h while six were selected for the other 2 groups. Area under the Receiver Operating Characteristic (AUROC) in the 3 groups was 0.82, 0.80, and 0.78, respectively, while sensitivity was 74%, 75%, and 70%, and specificity was 75%, 69%, and 70%, respectively. The SVM model was seen to perform better than other traditional screening tools such as the Berlin questionnaire, and the NoSAS Score [14].

Ramesh et al. used routinely acquired clinical data of 1479 records from the Wisconsin Sleep Cohort dataset. They found that the feature selection methods revealing the important primary predictors were waist-to-height ratio, waist circumference, neck circumference, body-mass index, lipid accumulation product, excessive daytime sleepiness, daily snoring frequency, and snoring volume. With a five-fold cross-validation strategy, Support vector machines achieved accuracy: 68.06%, sensitivity: 88.76%, specificity: 40.74%, F1-score: 75.96%, Positive predictive Value: 66.36% and Negative Predictive Value: 73.33% [9].

Bozkurt studies five different ML methods (Bayesian network, Decision Tree, Random Forest, Neural Networks, and Logistic Regression) in patients with clinical suspicion for OSA. They were able to show that the models performed well making use of clinical features, with a true positive rate as high as 0.71 and a true negative rate as low as 0.15. They were able to improve the accuracy of the classification models by including physical examination findings as features in the model [10].

Our group studied the use of an anthropometry-based model using ML and was able to show excellent sensitivity and predictive value [17].

Ghandeharioun et al. classified OSA severity using k-means and reported specificity and specificity and accuracy of 85.7%, 96.3%, 94%, respectively, with nonpolysomnographic features. However, no details were given regarding the training/testing strategy and OSA severity classifications’ results. In the validation phase, the 10-fold cross validated test results showed that Bayesian classification models are superior to other methods concerning classification accuracy in terms of minimum false positive rate (FPR), maximum true positive rate (TPR) and area under the ROC curve (*p* < 0.05). Although all the methods yield promising results for the classification of OSA, it is known that the performance of various classification algorithms strongly depended on the tested sample. Consequently, the superiority of Bayesian network over the other tested method holds for the given experiment only and cannot be generalized [18].

Sahin et al. retrospectively evaluated 390 patients that were referred to the sleep center. They used multi- variate linear regression analysis to identify independent AHI predictors and derived a formula AHI = ((0.797 × BMI) + (2.286 × NC) − (1.272 × SpO2)  +  (5.114 × TS)  +  (0.314 × WC)) to predict AHI. (WC = waist circumference, NC = neck circumference and TS = tonsil size). The authors found that up to 68.2% of the variation in the AHI was able to be explained with this formula [19].

In a study by Karamanlı et al. used an artificial neural network in the detection of OSA (AHI ≥ 10) based on patient demographic data (sex, age, BMI, and snoring status) and achieved high accuracy. Although the accuracy rate is reasonably high, they admitted that the model was tested with data from a small sample size [20].

Using a different approach to ML, Huo et al. recently described developing BASH-GN, a ML-based questionnaire to classify obstructive sleep apnea (OSA) risk by considering risk factor subtypes, using the SHHS test set (*n* = 1237) and Wisconsin Sleep Cohort (WSC) set (*n* = 1120). For purposes of comparison, they used four questionnaires that are commonly used to screen for OSA including the STOP-BANG, Epworth, and the Berlin. The model proved to be superior to these questionnaires on both test sets achieving AUROC (SHHS 1: 0.78, WSC: 0.76) and area under the precision-recall curve (AUPRC) (SHHS 1: 0.72, WSC: 0.74), respectively [15].

## 4. Knowledge Gaps

The performance of ML algorithms is primarily dependent on appropriate feature selection. One common theme that emerges is that models that include more objective based features perform better than those that include subjective based features based on patient self-reporting. Making use of different ML techniques to effect dimensionality reduction including principal component analysis, and singular value decomposition could help reveal weighted combinations of features that enhance the discriminating capacity of these models. Use of regularization and other techniques can help automate feature selection and bias inclusion of models with fewer parameters. More research may be needed to help improve feature selection using weighted combinations and automated selection in large population datasets. Furthermore, we may need to look at how factors like gender, ethnicity may influence feature selection.

We propose a three-stage tier system as an ideal model (Figure 2) to leverage the power of machine learning and apply it to a screening strategy for sleep apnea. This strategy may need to be prospectively validated in a large healthcare setting to evaluate its accuracy in detecting sleep apnea and measuring its ability to reduce healthcare resource utilization costs by reducing the burden of expensive testing for sleep apnea particularly involving in-lab polysomnography.

## 5. Limitations

There is important limitation to this review. Due to the limited scope of the review, we were unable to identify large numbers of articles specifically focused on use of ML to screen for risk or presence of sleep apnea. This clearly points to the need for many more research studies urgently needed in this area to evaluate the accuracy of one or more ML models and identify the ideal strategy in terms of feature selection and techniques to achieve the best results.

## 6. Conclusions

We are on the cusp of the next wave of approaches to better detect sleep apnea in the general population, with the more widespread use of ML. With the advent of wearable medical devices, we predicted that we would soon see a paradigm shift in the decision strategy for detecting sleep apnea. More reliance on objective measures such as body signal processing (i.e., heart rate or pulse waveforms, and anthropometric characteristics such as body weight, waist-hip ratio, neck circumference, and craniofacial and airway anatomy) are emerging. Smartphone-based apps that synthesize some or many of these features will be more prevalent. They could trigger patients to self-refer for more formal home sleep testing or polysomnography, thereby reducing the healthcare utilization of resources and improving the efficiency of diagnosing sleep apnea.

## Figures and Tables

**Figure 1 medicina-58-01574-f001:**
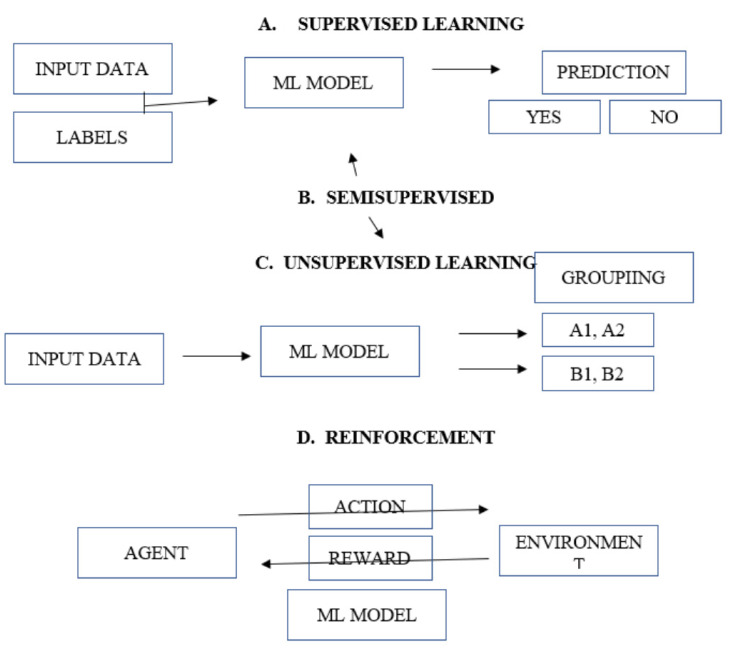
The four major machine learning models [1].

**Figure 2 medicina-58-01574-f002:**
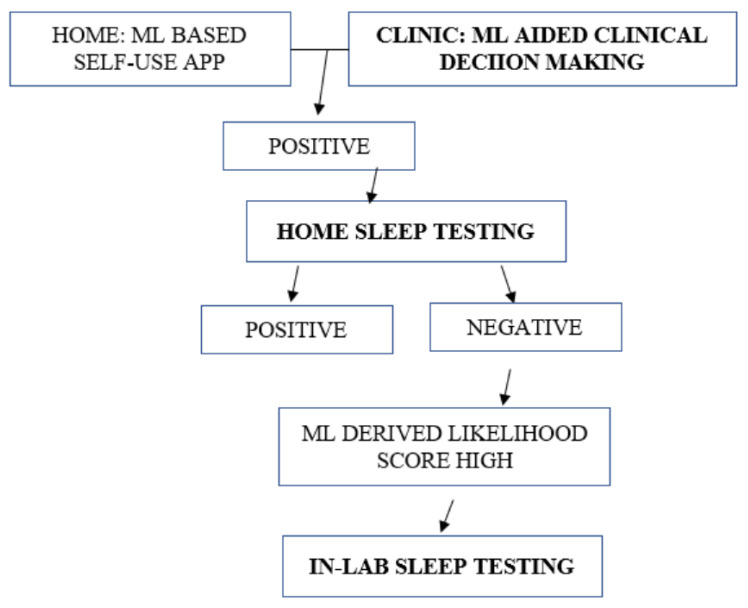
3 tier model for applying ml in screening for osa.

**Table 1 medicina-58-01574-t001:** Summary of major trials involving machine learning methods to predict OSA.

Study	Sample Size	Extractable Variables	Models Used	Methodology	Results
Mencar et al. [5]	313	19	7 Classification models #: 5 Regression models ##:	-trained classification models to predict the severity classes -trained regression models to directly predict the numerical AHI values.	**>Feature ranking** -BMI is the most important feature -FVC% and FEV_1_/FVC least important **>Classification models** -RF classification model trained with 1st 5 features -best results -CN trained with 6 features highest precision value **>Regression models** -SVM-*R* trained with 16 features is best predictive model for AHI
Ustun et al. [6]	1922	10	-SLIM model -7 State of the art classification models	SLIM and 7 classification methods to produce predictive models for OSA screening Features (i) self-reported symptoms; (ii) self-reported medical information	**Low prevalence** 64.2% sensitivity, 77% specificity LR^+^: 2.8, LR^−^: 0.46 Posttest probability patient w/OSA increase from 10% to 23%. w/out OSA decrease from 10% to 5% **High prevalence** -a negative result would lower\patient’s posttest OSA probability to 52%; not sufficient to have ruled out OSA.
Su et al. [7]	124	6	MMTS model	Multiclass MMTS based on anthropometric information and questionnaire data (ESS, SOS)	Accuracy of 84.38% OSA prediction, outperforming other methods including LR*, BPN, LVQ, SVM, C4.5 decision tree, and RS.
Zhang et al. [8]	481	7	SABIHC2 model	SABIHC2 model screens moderate to severe OSA based on faciocervical and anthropometric measurements	Better predictive ability than the STOP-BANG questionnaire. In asymptomatic patients (sensitivity (0.892 vs. 0.348) and specificity (0.755 vs. 0.809)) In patients experiencing sleepiness (ESS ≥10 (sensitivity (0.941 vs. 0.632) and specificity (0.740 vs. 0.727).
Ramesh et al. [9]	1479	8	7 Classification models+	56 continuous and categorical covariates are initially selected, the feature dimension narrowed systematically based on multiple feature selection methods according to their relative impacts on the models’ performance.	Performance with trained 19-EHR features -XGB model performs the best across the metrics of accuracy, sensitivity, F1-score, PPV, and NPV while LGBM still retains the highest specificity. Performance with trained 8-EHR features -SVM for classifying OSA patients at the cut-off of apnea-hypopnea index ≥5 and achieved accuracy: 68.06%, sensitivity: 88.76%, specificity: 40.74%, F1-score: 75.96%, PPV: 66.36% and NPV: 73.33%,
Bozkurt et al. [10]	338	14	5 machine learning method (Bayesian network, Decision Tree, Random Forest, Neural Networks and Multinomial Logistic Regression)	Classification of OSA severity of patients with suspected sleep disorder breathing as normal, mild, moderate, and severe based on (1) clinical data, (2) symptoms and (3) physical examination.	-Lateral Pharyngeal Wall Collapse (LPW) ranked 1 -Models achieved better performance using physical examination variables only but poorer performance using only clinical data and symptoms. -For OSA severity, Bayesian Network models showed the highest TPR, PPV, F Measure and AUC and lowest FPR and RMSE.
Kirby et al. [11]	255	23	Artificial neural networks (ANNs) to form a generalized regression neural network (GRNN) model	23 variables trained in training set—then the variables for the test set was presented to the trained network, and predictions were compared with actual outcomes.	Sensitivity for the diagnosis of OSA was 98.9% (95% CI, 96.7 to 100) Mean predictive accuracy of 91.3% (95% confidence interval (CI), 86.8 to 95.8). Sensitivity 98.9%, Specificity 80%, PPV88.1%, NPV 98%; AUC was 0.94.
Holf et al. [12]	17,448	4 age, sex, BMI, and race	LR and machine learning techniques, including artificial neural network (ANN), random forests (RF), and kernel support vector machine models	Patients were randomly split into training (*n* = 10,469) and validation (*n* = 6979) sets.	AUCs (95% CI) of ML models significantly higher than logistic regression (0.61 (0.60–0.62)) in both training and validation datasets (ANN, 0.68 (0.66–0.69); RF, 0.68 (0.67–0.70); and kernel support vector machine, 0.66 (0.65–0.67)). -OSA prediction tools using machine learning without patient-reported symptoms provide better diagnostic performance than logistic regression
Sun et al. [13]	120	29	Genetic algorithm (GA)	GA used to build 5 best models based on 110 validated questionnaires and Logistic regression model was used for comparison	Sensitivity of the GA models varied from 81.8% to 88.0%, with a specificity of 95% to 97%; vs. sensitivity and specificity of the LR model were 55.6% and 57.9%, respectively.
Huang et al. [14]	6875	2, 6, and 6 for AHI ≥5/h, ≥15/h, and ≥30/h, respectively.	SVM, LR, BQ (Berlin questionnaire), NoSAS, SLIM	Proposed a data mining-driven SVM model using a large-scale sleep lab-based data set to predict OSA with three different AHI cutoffs.	AUROC for AHI cutoffs, ≥5/h, ≥15/h, and ≥30/h were 0.82, 0.80, and 0.78, respectively. Sensitivty—74.14%, 75.18%, and 70.26%, Specificity −74.71%, 68.73%, and 70.30%, respectively. -Compared to logistic regression, Berlin questionnaire, NoSAS Score, and Supersparse Linear Integer Model (SLIM) scoring system, the SVM model performs better with a more balanced sensitivity and specificity.

Acronyms: SLIM: Supersparse Linear Integer Models; LR: likelihood ratio; MMTS: Mahalanobis-Taguchi system; SBP: systolic blood pressure, DBP: diastolic blood pressure; DI3: Frequency of desaturation (saturation index <3% in an hour); DI4: Frequency of desaturation (saturation index <4% in an hour); RDI: respiratory disturbance index, LR*: logistic regression, BPN: back propagation neural network, LVQ: learning vector quantization, SVM: support vector machine, C4.5 decision tree, and RS: rough set (RS); SABIHC2: Sex-Age-Body mass index (BMI)-maximum Interincisal distance-ratio of Height to thyrosternum distance-neck Circumference-waist Circumference; XGB: Extreme Gradient Boosting, LGBM: Light Gradient Boosting, CB: Catboost Algorithm, RF: Random Forest, SVM: Support Vector Machines. #: Majority vote, Naive Bayes, k-nearest neighbor, Classification tree, Random Forest (RF), Support vector machine (SVM), AdaBoost-SVM, CN2 rule induction. ##: Mean learner, LR, k-NN, Regression trees, Support vector regression (SVR), AdaBoost-SVR. +: XGB, LGBM, CB, RF, kNN, LR*, SVM. %- Bayesian network, Decision Tree, Random Forest, Neural Networks and Multinomial Logistic Regression.

## Data Availability

Not applicable.

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
