# Peer review of "Review of Application of Machine Learning as a Screening Tool for Diagnosis of Obstructive Sleep Apnea"

_medicina, 2022, doi:10.3390/medicina58111574_

Round 1
Reviewer 1 Report (Previous Reviewer 2)
Dear authors,
Thanks for your corrections.
Reviewer 2 Report (Previous Reviewer 1)
Figure 1 - A A. Supervised images - please erase A, there is double A
Otherwise, the paper raised in quality, all addressed points by previous reviewers were clarified. Well done.
This manuscript is a resubmission of an earlier submission. The following is a list of the peer review reports and author responses from that submission.
Round 1
Reviewer 1 Report
The authors presented a 7 pages long review article regarding machine learning in the diagnosis of obstructive sleep apnea with 15 references (including one reference to a recent article of their study group relevant to the topic). Overall, the main topic of the article would be promising. However, the article failed to follow the advised PRISMA guidelines for review articles on several points, did not involve important background information regarding the topic, did not include a detailed discussion and has multiple editing errors as well.
The article did not include key words.
In the introduction, the authors did not mention other diagnostic methods in the detection of OSA beside polysomnography and questionnaires.
In the second section called „Why machine learning?” the authors mentioned that there are various automated tools in the diagnose of OSA with two references, but failed to elaborate on the topic. They also mentioned four major machine learning models in this section and represented it in a figure as well. However, they only shortly explained two of the four models in the article.
The reference article (reference no 1) for Figure 1 (presenting four major models of machine learning) did not include any information regarding machine learning and it did not include the presented image either. The actual image is from the following article: Sarker, I.H. Machine Learning: Algorithms, Real-World Applications and Research Directions. SN COMPUT. SCI. 2, 160 (2021). https://doi.org/10.1007/s42979-021-00592-x (Figure 2) which is not cited in this current article at all. This could lead to copyright issues.
The authors failed to specify any inclusion and exclusion criteria for the studies analysed in the article. The article did not specify any search strategies, database, registers or other sources searched or consulted to identify studies for the article.
The authors properly described the studies included in the article. However, based on my short research there are several recently published studies and reviews that would have been worth mentioning as well. Just to list a few:
https://doi.org/10.1007/s11325-021-02301-7
https://doi.org/10.2147/NSS.S297856
https://doi.org/10.3390/healthcare9070914
https://doi.org/10.1038/s41598-020-62223-4
https://doi.org/10.3389/fnins.2022.726880
https://doi.org/10.1016/j.chest.2021.10.023
https://doi.org/10.1186/s40463-022-00566-w
https://doi.org/10.1088/1742-6596/1937/1/012054
https://doi.org/10.1145/3433987
The authors also failed to deliver a detailed discussion regarding the results, limitations or specific conclusions of their review. They did not identify knowledge gaps that may be potential topics for future research.
The numbering of the references in the main text is not in ascending order (after reference “2”, reference “14,15” follows).
In the main text, the authors failed to define several abbreviations. They also did not use some abbreviations consistently (such as 'OSA' and 'OSAS').
The Author Contribution, Funding, Institutional Review Board Statement, Informed Consent Statement, Data Availability Statement and Conflicts of Interest Statement are missing.
Reviewer 2 Report
Dear Authors,
Thanks for your manuscript.
Your study is not resource rich. The review of the studies has not been done well. At least 30 reliable sources should be used for a review study. Only 15 sources have been used in this study. In my opinion, this study is not suitable for publication in this journal.
Reviewer 3 Report
Overall, it is a good review article about AI for diagnosing OSA.
I would recommend ensuring the abbreviations are referenced clearly.
I think in line 156, you meant sensitivity and specificity.
Good luck!